# Study of the Structure of FeO*x*-CaO-SiO₂-MgO and FeO*x*-CaO-SiO₂-MgO-Cu₂O-PdO Slags Relevant to Urban Ores Processing through Cu Smelting

**Mohammad Mehedi Hasan** [1,*], **M. Akbar Rhamdhani** [1] , **M. Al Hossaini Shuva** [2] **and Geoffrey A. Brooks** [1]

1   Fluid and Process Dynamics (FPD) Group, Swinburne University of Technology, Melbourne 3122, Australia; arhamdhani@swin.edu.au (M.A.R.); gbrooks@swin.edu.au (G.A.B.)
2   PF Metals Resource Recovery/Envirostream, Melbourne 3101, Australia; mahshuva@gmail.com
*   Correspondence: mohammadhasan@swin.edu.au; Tel.: +61-4-2021-9123

**Abstract:** Ferrous-calcium-silicate (commonly known as FCS) slags are used in the valuable metal recycling from urban ores through both primary and secondary copper smelting processes. In the present study, the structure of selected FCS-MgO (FCSM) and FCS-MgO-Cu₂O-PdO (FCSM-Cu₂O-PdO) slags, relevant to the processes, were investigated using Fourier-transform infrared (FTIR) spectrometry. Deconvolution of the FTIR spectra was carried out to calculate the relative abundance of different silicate structural units ($Q^n$), the overall degree of polymerization (DOP) of the slags and the oxygen speciation in the FCS slags. It was observed that, for the slag investigated, the relative intensity of both the high-frequency band $\approx 1100$ cm$^{-1}$ ($Q^3$) and low-frequency band $\approx 850$ cm$^{-1}$ ($Q^0$) were affected by Fe/SiO₂ ratio, basicity, temperature (T) and oxygen partial pressure ($pO_2$). The DOP and the average number of bridging oxygen (BO) were found to decrease with increasing both Fe/SiO₂ ratio and basicity. Improved semi-empirical equations were developed to relate the DOP of the slags with chemistry, process parameters and partitioning ratio (i.e., the ratio of the amount of element in the slag phase to metal phase, also known as distribution ratio) of Pd and Ge. Possible reactions, expressed as reactions between metal cations and silicate species, as a way to evaluate thermodynamic properties, are presented herein.

**Keywords:** ferrous-calcium-silicate (FCS) slag; black copper smelting; slag structure; FTIR spectrometry; E-waste; urban ores; palladium; germanium

## 1. Introduction

Slags play a vital role in pyrometallurgical processes, such as in the production, recycling and refining of liquid metal. The chemistry and structure of a slag controls the removal of impurities from the liquid metal to the slag, and vice versa—the partitioning of the selected valuable elements to metal phase from the slag [1,2]. The slag chemistry also influences both the thermophysical and thermodynamic properties, such as density, viscosity, electrical conductivity, foaming index, thermal conductivity, surface tension, partition ratio, molar entropy, diffusivity and mixing free energy of silicates [3–5]. Successful operations of metal productions are dependent on these properties. Therefore, a clear understanding of the relations between the slags' chemistry, structure and their properties is vital for designing suitable slags for the appropriate process conditions.

Urban ores are the waste materials that are sources of many base metals and other valuable elements. Urban ores consist of electronic waste (E-waste), low grade copper tailings, industrial sludge, spent catalysts and other low-grade scraps in metallic and oxidized states. In the context of processing

of urban ores (such as E-waste, which contains Au, Ag, Pt, In, Ru, etc.) through black copper smelting, understanding the inter-relation between the slag's chemistry, structure and properties is important for maximizing the recovery of the valuable elements into the copper phase.

Metallurgical slags are composed of different oxide components that can be acidic (e.g., silicate), basic (e.g., calcia) or amphoteric (e.g., alumina, that can behave as both acid and base depending on the conditions). Generally, acidic oxides are network formers; their addition increases the number of bridging oxygen ($O^0$) in silicate melts. Basic oxides, on the other hand, are network breakers that produce free oxygen ions ($O^{2-}$) and non-bridging oxygens ($O^-$) at the expense of bridging oxygen ($O^0$) in the silicate melt. This can be expressed in Equation (1) [6]:

$$O^0 + O^{2-} = 2O^-. \tag{1}$$

Silicate units in the melt are formed by the polymerization of $[SiO_4]^{4-}$ tetrahedrons. Virgo et al. [7] identified different anionic structural units according to the number of non-bridging oxygens (NBO) present in the tetrahedral units from Raman and FTIR spectra deconvolution. They expressed different anionic units using $Q^n$ notation, where n is the number of bridging oxygens. The $Q^0$ ($[SiO_4]^{4-}$), $Q^1$ ($[Si_2O_7]^{6-}$), $Q^2$ ($[SiO_3]^{2-}$), $Q^3$ ($[Si_2O_5]^{2-}$) and $Q^4$ ($SiO_2$) are, respectively, the monomer, dimer, chain/ring and 3D-network of silicate tetrahedrons formed by the increasing number of bridging oxygens at the corner of the silicate tetrahedral units. The term degree of polymerization (DOP), which corresponds to the fraction of highly polymerized structural unit $Q^4$ in the silicates, has been commonly used and is measured by the parameter $Q^3/Q^2$ [4]. Another structural parameter usually used to represent the silicate structure is non-bridging oxygen per tetrahedral units (NBO/T), which is calculated from the relative abundance of different structural units, as shown in Equations (2) and (3):

$$(BO/T) = 4 - (NBO/T); \tag{2}$$

$$NBO/T = (4 \times Q^0) + (3 \times Q^1) + (2 \times Q^2) + (1 \times Q^3) + (0 \times Q^4). \tag{3}$$

There have been many studies carried out investigating the structure of silicate melts in metallurgical applications; however, the focus has been on slags with compositions relevant to the ironmaking and steelmaking applications, and there are no published studies that focus on copper-making slags.

Previous research suggests that printed circuit boards (PCBs) contain as much as 220 ppm of Pd [8], which is equivalent to USD 13,747 per ton of circuit boards [9]. As PCBs are rich in copper and other valuable metals, secondary copper smelters, including Boliden Ronnskar, Umicore, Dowa mining, L.S. Nikko and others, use PCBs as the raw materials in the process. The inclusion of PdO in the slag is expected during the smelting process, as PCBs contain Pd. A thorough study on the effect of process parameters and slag compositions on the partitioning of Pd in the process is necessary to ensure maximum recovery of the metals. Moreover, the effect of PdO on the structure of slag is also important, as the thermophysical and thermodynamic properties of the slags are related to slag structures [10].

Traditionally, iron silicate ($FeO_x$-$SiO_2$) slags are used for copper smelting processes, but this slag system, if not properly controlled, has the problem associated with magnetite precipitation, which is not good for the operation [11]. This slag system generally exhibits a high viscosity and possesses the risk of formation of slag foaming during the operation. A different slag system of calcium ferrite ($CaO$-$FeO_x$) was first introduced for continuous copper processing by Mitsubishi. This highly basic slag overcomes the limitation of the iron silicate slag, but it aggressively attacks the refractories, resulting in significant wear-off the furnace linings. Currently, $FeO_x$-$CaO$-$SiO_2$ (FCS) slags with typical compositions of 45–65 wt.% $FeO_x$, 3–10 wt.% CaO and 30–45 wt.% $SiO_2$ are used for industrial copper smelting, which has intermediary basicity to that of iron silicate and calcium ferrite slags and overcomes the limitation of these two slags [12,13]. Magnesia-chrome refractory lining is used as a wear protection layer in

copper production furnaces. The presence of MgO in the FCS is considered to simulate a situation in the industrial copper processing where dissolution of the MgO-composite refractory into the slag occurs. Therefore, the study of the structure of FCS-MgO (FCSM) slags and its effect on the partitioning of the valuable elements is necessary.

In the current work, the structures of FCSM and FCS-MgO-Cu$_2$O-PdO (FCSM-Cu$_2$O-PdO) slags relevant to black copper smelting were studied. The effects of the slag composition, partial pressure of oxygen ($pO_2$) and temperature on DOP of the FCS-based slags were investigated. Improved semi-empirical relationships between the process parameters (chemistry, temperature and partial pressure of oxygen) and DOP of the melt were developed, including correlations that include the partitioning ratio of Pd and Ge.

## 2. Previous Studies on Structures of Copper-Making Slags

Kaur et al. [14,15] investigated the wear of a magnesia-chrome furnace lining by different copper smelting slags at high temperatures and showed that the FCS slag is less corrosive to the refractories. They also investigated the partition ratios of Ni, Pb and Sb between the slag and copper and reported similar values compared to that of calcium ferrite slag. A number of researchers recently investigated the thermodynamics and partition of various valuable elements between liquid copper and liquid slag in the context of processing of electronic waste through secondary copper smelting (black copper smelting); i.e., Bi [16], Sn and In [17–20]; Ga [20]; and Ge, Ta and Pd [20–22]. It has been reported that there are only limited partition ratio data available in the open literature, particularly that are relevant to secondary copper (black copper) smelting operations [23]. There is a limited work published in the open literature on the investigation of the interrelationships between composition, structure and properties of MgO-containing FCS slags relevant to primary and secondary copper-making.

In the case of ironmaking and steelmaking slags, Jiao et al. [24] investigated the correlation between the electrical conductivity and slag composition at different temperatures in the SiO$_2$-CaO-MnO-MgO slags. They concluded that the addition of basic oxides, such as CaO, MgO, FeO and MnO, increases the electrical conductivity, whereas addition of acidic oxides, such as SiO$_2$ and P$_2$O$_5$, gives a decreasing effect. Lee et al. [25] studied the effect of FeO and MgO on the viscosity of CaO-SiO$_2$-Al$_2$O$_3$ (10–13 wt.%), MgO (5–10 wt.%) and FeO (0–20 wt.%) slags. They explained the decrease of viscosity with increasing amount of FeO and MgO by the increasing of slag depolymerisation. Kim et al. [26] observed a decreasing trend of a foaming index with addition of FeO (up to 20 wt.%) in the CaO-SiO$_2$-FeO-Al$_2$O$_3$ slag; however, addition of FeO beyond this did not decrease the foaming index any further. The effect of Al$_2$O$_3$ on foaming index was not clearly revealed in the study.

In the above-mentioned works, direct measurements of the structure of the slags were not carried out. In other study, McMillan [27] prepared CaO-MgO-SiO$_2$ melts using solar furnace technique and measured the distribution of different silicate structural units in the melts. McMillan observed the decrease of polymerization of the melts as the silica content was decreased and reported that the Mg substitution of Ca reduced the $Q^3/Q^2$. Brandriss and Stebbins [28] investigated the effect of temperature on the structure of slags CaO (25 wt.%), MgO (25 wt.%), SiO$_2$ (50 wt.%) and other silicate systems. They concluded that the increase of temperature changes the $Q^n$ speciation and this substantially accounts for the difference of the total enthalpy of the glass and crystal. Cooney et al. [29] also observed and reported a similar network breaking action of Mg$^{2+}$ and Ca$^{2+}$ ions. Sohn and Min [30] collated the viscosity data of the CaO-SiO$_2$-Al$_2$O$_3$-MgO systems and showed that the addition of MgO reduces the viscosity of the melt. The general role of MgO is to break the silicate networks; however, in highly basic slag the behavior of MgO can deviate. Mg$^{2+}$ may behave as a charge-balancing ion if the slag contains a significant amount of Al$_2$O$_3$. The structure and viscosity of SiO$_2$-CaO-MgO-Al$_2$O$_3$ (9 wt.%) were studied by Gao et al. [31]. They reported the transformation of the slag to a simpler structure with increasing basicity or MgO, and therefore, the viscosity and the activation energy of viscous flow also decreased.

Very limited studies have been carried out correlating the slag structure with the thermodynamic parameters in the relevant slag systems. Park [4] investigated the relationship between sulfide capacity and structure of CaO-SiO$_2$-MnO and CaO-SiO$_2$-MgO slags and obtained a relationship between sulfide capacity and non-bridging oxygen. In that research, it was revealed that the excess free energy of oxides in slag (CaO, MgO, MnO) is strongly dependent on the abundance of silicon anionic units, which was also elucidated further with ionization potential ($Z/r^2$) of the metal ions. In another study, Park et al. [32] reported an increase of depolymerization of the CaF$_2$-CaO-SiO$_2$-FeO$_x$ slag system with increasing FeO$_x$ (0 to 21.5 wt.%). In this basic slag, FeO$_x$ acted as a polymerizing agent of the silicate slag. They also reported an increase in depolymerization due to the reaction of hydrogen with bridged oxygen in acidic slag. Hydrogen produces free hydroxyls by reacting with free oxygens and increases the DOP in highly basic slags.

Recently, the author's group studied the relation of slag structure with thermodynamics of Ge at black copper smelting conditions [33]. A semi-empirical equation relating thermodynamics (partition ratio of Ge in slag and copper, $L_{Ge}^{\frac{s}{m}}$) with slag structure ($Q^3/Q^2$) and process parameters (temperature and partial pressure of oxygen) was developed. In the case of Pd, there are only limited studies which focus only on the Pd partition between slag and matte under primary copper production conditions (matte-slag system) [34–36]; and no published works on the effect of slag structure on Pd-partition ($L_{Pd}^{\frac{s}{m}}$). Recently, a conference publication by the authors who wrote [37] explained some observations of the effects of Fe/SiO$_2$ ratio and basicity on the structure of Pd and Ge-containing slags. In this current work, an improved deconvolution approach for analyzing the FTIR data was adopted using more complete and comprehensive data. This resulted in improved semi empirical equations that relate partition ratio of Ge and Pd in the slag and copper with the slag structure and process parameters.

In summary, there have been previous studies in the slag system FeO$_x$-CaO-SiO$_2$-MO, as described above. Nevertheless, there is still lack of information on the structures of FCS slags, particularly regarding slag composition and conditions applicable to secondary (black) copper smelting practice (high concentration of FeO$_x$ and SiO$_2$, and at low oxygen potential) for recycling of valuable metals from urban ores, such as E-waste.

## 3. Methodology of Experiments

### 3.1. Materials, Master Alloys and Master Slags

The raw materials used for the master alloys and master slags production were obtained from Alfa Aesar; i.e., silicon (IV) oxide (99.5 wt.%); calcium carbonate (99.5 wt.%); iron (>99.8 wt.%); iron (III) oxide (99.99 wt.%) powders; and copper flakes (>99.99 wt.%). Palladium powder (99.999 pct) and palladium oxide (99.99 pct), obtained from Sigma-Aldrich, were also used in the study.

For the making of master FCSM slag, the CaCO$_3$ powder was firstly dried to remove moisture. This was carried out at 500 °C for 5 h; that was followed by calcination at 1100 °C for 4 h in an alumina tray using a muffle furnace. Powders of the appropriate composition were mixed and ball milled for a period of 36–48 h. The mixed powder samples were then pre-melted at 1300 °C and at $pO_2 = 10^{-8}$ atm in a vertical resistant-tube furnace that use MoSi$_2$ heating elements (Nabertherm, Lilienthal, Germany). High purity magnesia crucibles (97.4% MgO, Ozark Technical Ceramics, Webb City, MO, USA) were used for melting the samples. The master slags were remelted at least two times to ensure uniformity and homogeneity of the slag compositions. The compositions of slags were checked by bulk chemical composition analyses using ICP-AES (inductively-coupled-plasma atomic-emission-spectroscopy technique). The slag preparation procedure is shown in Figure 1.

The Cu-Pd master-alloy was prepared from copper flakes and palladium powders. Copper flakes were mixed with Pd (6 wt.%) and equilibrated for 6 h. This was carried out in an MgO crucible inside a vertical tube furnace. The composition of the master alloy was confirmed using ICP-AES technique; and less than 0.1 wt.% of palladium was lost from the system during the melting.

The FCSM-Cu$_2$O-PdO slags were prepared by equilibration of the FCSM slag with the Cu-Pd master-alloy for 16 h. The thermodynamics of the system at the given temperature and partial pressure of oxygen allowed partial oxidation of the master-alloy and contributing the Cu$_2$O and PdO species to the slag.

High purity gases used for the experiments were supplied by Coregas. High purity argon gas was passed through a gas cleaning system before entering the furnace to ensure inside atmosphere has a low moisture and oxygen content. To control the oxygen partial pressure at a certain temperature, a mixture of high-purity carbon dioxide and carbon monoxide gases was used. The details of the gases used are given as follows: carbon dioxide (CO$_2$), 99.995 pct purity, grade 4.5; carbon monoxide (CO), 99.5 pct purity, grade 2.5, the major impurities being nitrogen and carbon dioxide; and Argon (Ar), 99.995 pct purity, grade 5.

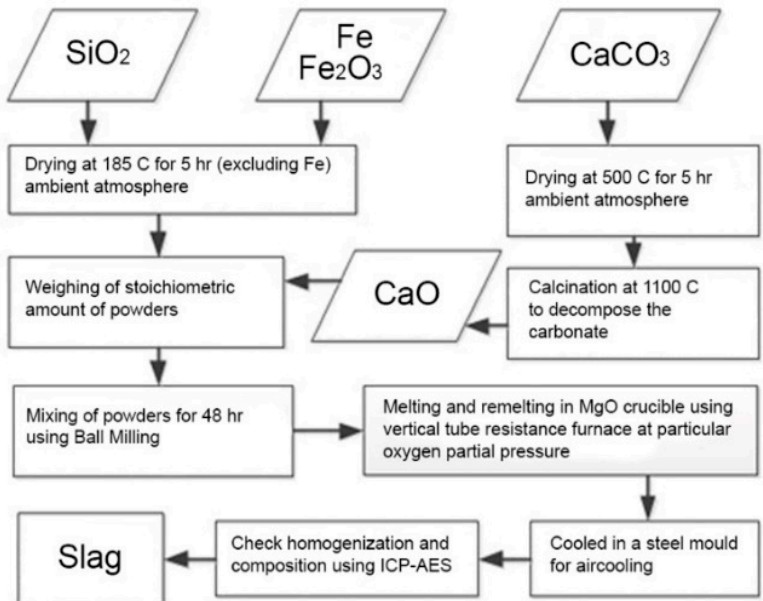

**Figure 1.** Master slag preparation procedure, melting carried out at 1300 °C and at $p$O$_2$ (oxygen partial pressure) of $10^{-8}$ atm.

## 3.2. Experimental Apparatus and Plan

A R-type (Pt/Pt-13 wt.% Rh) reference thermocouple was used to calibrate the working thermocouples prior to the experiments. After purging the furnace with high purity argon, carbon dioxide and carbon monoxide were mixed and allowed to flow into the furnace to target a particular $p$O$_2$ with a total flowrate of 400 mL/min. Digital mass flow controllers (from Aalborg, Chicago, IL, USA) were used to control the CO$_2$ and CO gas flows. The gas mixture was passed through the furnace gas inlet at the bottom to provide a good gas flow around the samples and minimize the effect of thermal segregation. A gas bubbler was connected to exit gas line to avoid oxygen back diffusion, e.g., from the air back into the furnace. The required flow rate ($R$) of each individual gas was calculated from the relationship suggested by Yazawa and Takeda [38], as shown in Equation (4).

$$\log p\mathrm{O}_2 = 2\log\!\left(\frac{R_{CO_2}}{R_{CO}}\right) - \frac{29510}{T} + 9.05, \tag{4}$$

where $R = R_{CO2} + R_{CO} = 400$ mL/min.

The slag equilibration experiments were carried out inside the vertical tube resistance furnace, which is shown schematically in Figure 2 [22]. The slag was placed in a magnesia crucible and allowed to equilibrate with the atmosphere (of particular oxygen partial pressure) at the target temperatures.

An oxygen probe, i.e., a DS-type SIRO$_2$ C700+ solid zirconia electrolyte oxygen sensor, was used to measure and confirm the $pO_2$ inside the furnace. The thermocouple was supplied by Ceramic Oxide Fabricators Australia.

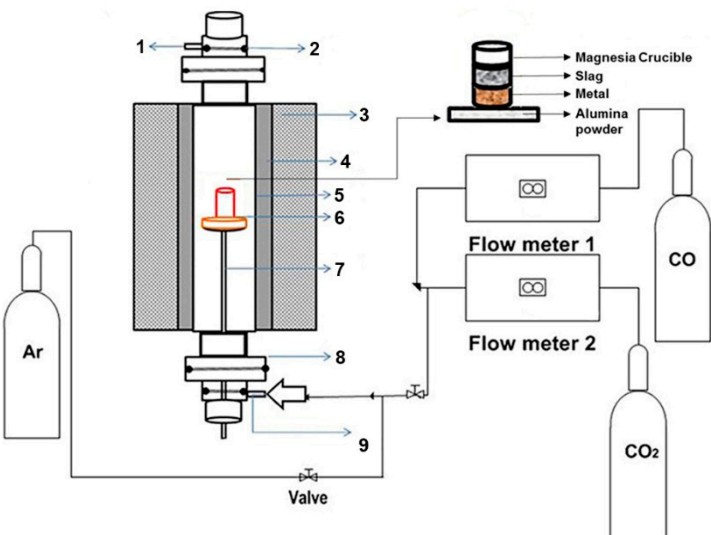

**Figure 2.** A schematic of experimental apparatus consisting of a vertical tube furnace, digital gas flow meters and different gases used in the current study. Legend: 1—gas outlet, 2—silicone O-ring, 3—MoSi$_2$ heating elements, 4—alumina tube, 5—mullite tube, 6—magnesia crucible, 7—alumina pedestal, 8—water-cooled flange, 9—gas inlet [22].

For the study on the FCSM slag, the effect of different slag compositions on the structure was investigated by varying the basicity and the Fe/SiO$_2$ ratio. The basicity in the current study was defined as B = [wt.% CaO + wt.% MgO]/wt.% SiO$_2$ and was varied between 0.49 to 0.89. The Fe/SiO$_2$ ratio of the slag was varied between 0.81 to 1.16. The temperature and $pO_2$ were fixed at 1300 °C (1573 K) and $10^{-8}$ atm.

For the study using the FCSM-Cu$_2$O-PdO slag, the temperature was varied between 1200 and 1350 °C (1473 to 1623 K), $pO_2$ between $10^{-8}$ to $10^{-10}$ atm, Fe/SiO$_2$ ratio between 0.8 to 1.2, and basicity between 0.34 to 0.65. The main experiments involved equilibrations between the master slags and the Cu-Pd master alloys. The equilibriums were approached from both reduced and oxidized conditions. Master slag (3.5 g), copper (2.8 g) and master alloy (Cu-5.9 wt.% Pd) (0.7 g) were used for equilibration experiments approached from reduced condition. For the experiments from oxidized conditions, the following were used; i.e., copper (3.5 g), master slag (3.45 g) and PdO (0.05 g). The details of the preparation of the FCSM-Cu$_2$O-GeO$_2$ slag and equilibration experiments are mentioned in another article [21].

Several initial experiments were carried out at $pO_2 = 10^{-8}$ atm to determine the appropriate minimum equilibration-time for the FCSM-Cu$_2$O-PdO slag [22]. These experiments were conducted using a slag with Fe/SiO$_2$ ratio of 0.99 and 11.5 wt.% of CaO slag, and copper, at 1300 °C (1573 K). The equilibria were approached both from reducing and oxidizing compositions. The results of chemical analysis of slag and copper phase from the equilibrium experiments shows that the palladium partition ratio $L_{Pd}^{\frac{s}{m}}$ and copper loss to slag are converged to equilibrium conditions after 16 h of equilibration. Considering these results, an equilibration time of 16 h was used for all the experiments with Pd-containing slag as it deemed to be sufficient to allow proper equilibration. The results of these measurements have been presented elsewhere [22].

In addition to the FTIR analyses, the partition ratio of Pd, $L_{Pd}^{\frac{s}{m}}$, was also measured. The summary of the experimental conditions investigated in the current work for the studies of FCSM and

FCSM-$Cu_2O$-PdO slags is given in Table 1. It should be noted that these values of compositions were analysed using ICP-AES.

**Table 1.** Experimental conditions and the equilibrium compositions of the FCSM and FCSM-$Cu_2O$-PdO slags.

| Temperature °C (K) | $pO_2$ (atm) | Slag Composition (wt.%) | | | | | | $Fe_T/SiO_2$ | Basicity | Samples Group No. |
|---|---|---|---|---|---|---|---|---|---|---|
| | | $Fe_T$ | $SiO_2$ | CaO | MgO | $Cu_2O$ | PdO (ppm) | | | |
| | | | | | FCSM slags | | | | | |
| | | 27.4 | 33.9 | 9.7 | 6.8 | - | - | 0.81 | 0.49 | #1 |
| | | 32.8 | 33.1 | 9.9 | 6.5 | - | - | 0.99 | 0.49 | #2 |
| 1300 (1573) | $10^{-8}$ | 37.9 | 32.8 | 9.8 | 6.5 | - | - | 1.16 | 0.49 | #3 |
| | | 33.4 | 33.6 | 14.5 | 7.5 | - | - | 0.99 | 0.65 | #4 |
| | | 30.7 | 31.3 | 19.4 | 8.8 | - | - | 0.99 | 0.89 | #5 |
| | | | | | FCSM-$Cu_2O$-PdO slags | | | | | |
| | $10^{-8}$ | | | | | 1.06 | 592 | | | #4-a |
| 1300 (1573) | $10^{-9}$ | 31.1 | 32.6 | 14.5 | 7.8 | 1.80 | 99 | 0.95 | 0.68 | #4-b |
| | $10^{-10}$ | | | | | 2.56 | 29 | | | #4-c |
| 1200 (1473) | | | | | | 2.82 | 1397 | | | #2-a |
| 1250 (1523) | $10^{-8}$ | 32.8 | 33.1 | 9.9 | 6.5 | 1.84 | 1131 | 0.99 | 0.49 | #2-b |
| 1350 (1623) | | | | | | 1.13 | 282 | | | #2-d |
| | | 37.3 | 35.3 | 5.6 | 6.3 | 1.61 | 1061 | 1.06 | 0.34 | #6 |
| | | 33.4 | 33.6 | 14.5 | 7.5 | 1.65 | 353 | 0.99 | 0.65 | #4-d |
| 1300 (1573) | $10^{-8}$ | 27.4 | 33.9 | 9.7 | 6.8 | 1.55 | 11,219 | 0.81 | 0.49 | #1-a |
| | | 32.8 | 33.1 | 9.9 | 6.5 | 1.61 | 661 | 0.99 | 0.49 | #2-c |
| | | 37.9 | 32.8 | 9.8 | 6.5 | 1.52 | 50 | 1.16 | 0.49 | #3-a |

The main experiments were terminated by flowing high purity Ar gas. The crucible containing the sample was then lowered to the cool zone at the bottom of the furnace for quenching. The crucible system was then collected and carefully crushed and separated for further analyses.

### 3.3. Slag and Metal Samples Characterization

The obtained slag samples were broken into pieces, and then ground to powders. The bulk compositions of the slags and metals were determined and reconfirmed using ICP-AES. A finely ground slag sample was fused in sodium peroxide at high temperature; then, the mixture was diluted with nitric acid. For metal analyses; the samples were cut into small pieces and then dissolved in hydrochloric acid which was further diluted in distilled water. The solution was transferred to a volumetric flask for the analysis of copper and palladium concentrations in the solution against multiple standard solutions. Samples were analysed for concentration of calcia, total iron, silica, magnesia and palladium.

A Nicolet iD5 attenuated total reflection FTIR spectrometer (Thermo Fisher Scientific, Waltham, Massachusetts, USA), equipped with a diamond crystal (with an incident angle of 42°) and ZnSe lens, was used in the current study. The spectra of the slag were taken from the 200–1600 $cm^{-1}$ frequency range (with 100 scans at a resolution of 4.0 $cm^{-1}$) and the measurements were carried out at room temperature. OMNIC software (OMNIC 8.0, Thermo Fisher Scientific, Waltham, MA, USA) was used for recording the spectra. The spectra were deconvoluted using Gaussian functions using the CasaXPS software (Version 2.3.19PR1.0, Casa Software Ltd., Teignmouth, Devon, UK). This provided more accurate results compared to the previous work [33]. The relative abundances of the $Q^n$ units were determined by calculating the area fractions of the fitted Gaussian curves at different silicate unit frequencies. The frequencies (wavenumbers) of the anionic silicate units used in the procedure are given in the Table 2.

**Table 2.** FTIR/Raman shift with vibration mode for different silicate structure units.

| NBO/T ($Q^n$) | Unit | Frequency (cm$^{-1}$) | Mode | Ref. |
|---|---|---|---|---|
| 0 ($Q^4$) | $SiO_2$ | 1200, 1190 | Asymmetric stretch | [7] |
| 1 ($Q^3$) | $[Si_2O_5]^{2-}$ | 1100–1050 | Symmetric stretch | [7] |
| 2 ($Q^2$) | $[SiO_3]^{2-}$ | 980–950 | Symmetric stretch | [7] |
| 3($Q^1$) | $[Si_2O_7]^{6-}$ | 920–900 | Symmetric stretch | [7] |
| 4 ($Q^0$) | $[SiO_4]^{4-}$ | 880–850 | Symmetric stretch | [7] |
| - | Si–O–Si | 800–780 | Asymmetric bending | [27] |
| - | Si–O–Si | 650–500 | Symmetric bending | [27] |
| - | Si–O–Si | 480–440 | Rocking | [27] |

It is worth mentioning here that the structural analyses were carried out on silicate slag samples that were quenched from the equilibration temperature to room temperature. It has been shown by previous investigators that for silicate slags, there is no significance difference between the spectra obtained from quenched ones and those of in-situ slags at high temperature [39,40].

## 4. Results and Discussion

### 4.1. Structure Analysis of the FeOx-CaO-SiO$_2$-MgO (FCSM) Slags

#### 4.1.1. The Effect of Fe/SiO$_2$

The FTIR spectra of the FCSM slag samples with Fe/SiO$_2$ ratios from 0.81 to 1.16 are shown in Figure 3a. The characteristic FTIR spectra of the silicate anionic units are typically found in the wavenumber range of 700 cm$^{-1}$ to 1200 cm$^{-1}$. The positions of the Si-O-Si bands in the tetrahedral network are dependent on the bridging condition of the oxygen atoms in the tetrahedrons [7]. Figure 3a shows the peak shifting towards low wavenumber for $Q^3$ and $Q^0$ with increasing Fe/SiO$_2$ ratio. The intensity decrease for the high frequency absorbance band $Q^3$ indicates the breakdown or depolymerization of the tetrahedral network in the slag [38–40]. The peak shifting in the FTIR spectra is also an indication of an increase in the number of non-bridging oxygen per silicon tetrahedral (NBO/T). The increase of NBO/T in silicate melts also support the notion that the depolymerization of slag increases with increasing Fe/SiO$_2$ ratio. The deconvoluted FTIR spectra of the FCSM slag samples with different Fe/SiO$_2$ are given in Figure 3b–d. The peak positions and relative abundances of the peaks corresponding to different structural units are listed in Table 3. The envelope of $Q^2$ and $Q^3$ silicate units in the FCSM slag were observed to decrease with increasing Fe/SiO$_2$.

**Table 3.** FTIR peak shifts and area fractions from FTIR spectra of FCSM slags with different Fe/SiO$_2$ ratios.

| Sample | Fe/SiO$_2$ | Peak Position (cm$^{-1}$) | | | | Fraction of Different $Q^n$ Units | | | | $Q^3/Q^2$ |
|---|---|---|---|---|---|---|---|---|---|---|
| | | $Q^0$ | $Q^1$ | $Q^2$ | $Q^3$ | $Q^0$ | $Q^1$ | $Q^2$ | $Q^3$ | |
| #1 | 0.81 | 870 | 920 | 1007 | 1091 | 0.26 | 0.42 | 0.18 | 0.15 | 0.86 |
| #2 | 0.99 | 852 | 919 | 1011 | 1109 | 0.24 | 0.42 | 0.24 | 0.10 | 0.42 |
| #3 | 1.16 | 849 | 914 | 998 | 1097 | 0.36 | 0.44 | 0.13 | 0.07 | 0.57 |

The fractions of the silicate units obtained from the FTIR spectra were plotted as a function of Fe/SiO$_2$ and they are shown in Figure 4. It was observed that the $Q^1$ is the major unit in the slag. The $Q^0$ fraction was found to be unchanged with increasing Fe/SiO$_2$ from 0.81 to 0.99. In this range, it appeared that the structure mainly transformed from highly polymerized $Q^3$ unit into depolymerized $Q^2$ unit as Fe/SiO$_2$ increased from 0.81 to 0.99. It was observed that the fraction of $Q^0$ unit increased significantly as the Fe/SiO$_2$ ratio increased from 0.99 to 1.16, while $Q^3$ and $Q^2$ units decreased. That indicates the depolymerization of the silicate structure through transformation from both highly polymerized $Q^3$ and $Q^2$ into $Q^0$ silicate units at Fe/SiO$_2$ beyond 0.99.

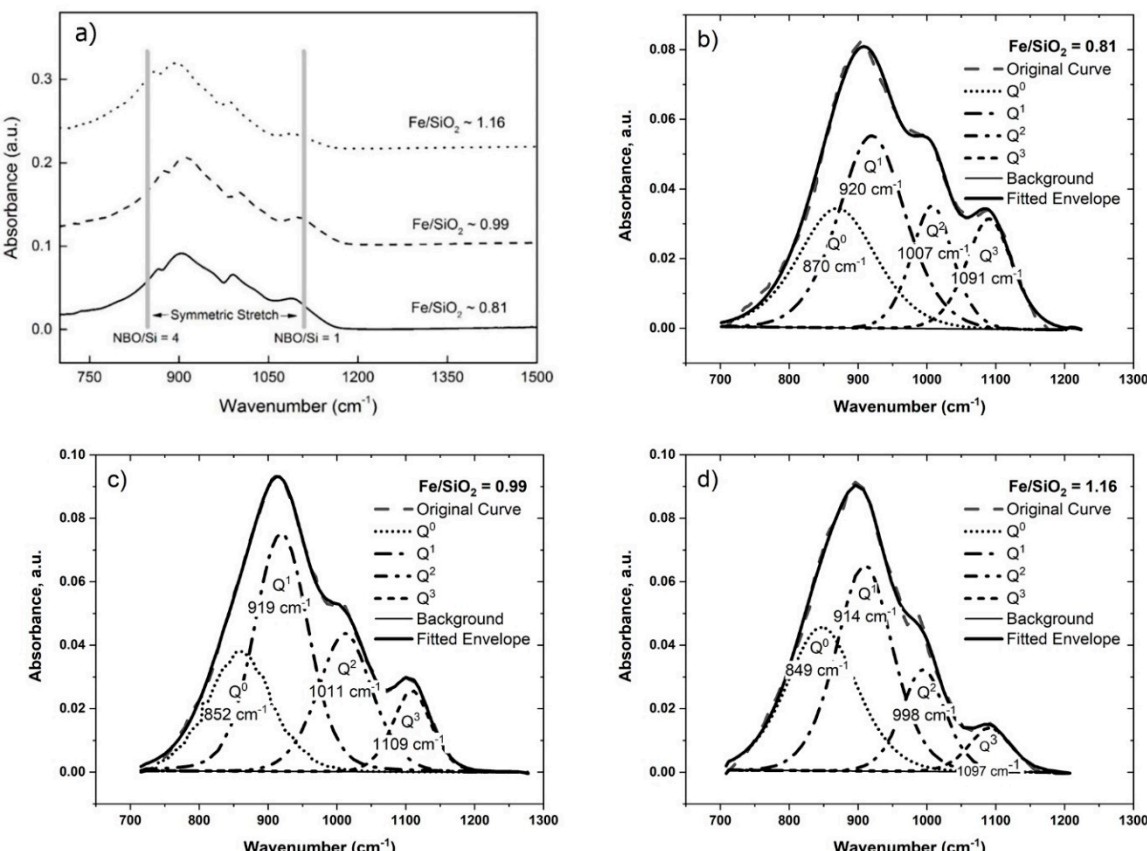

**Figure 3.** (**a**) FTIR spectra of FCSM slags containing $Fe/SiO_2$ ratios from 0.81–1.16 ($pO_2 = 10^{-8}$ atm and $T = 1300\ °C$); and deconvolution results of FTIR spectra for slags with (**b**) $Fe/SiO_2 = 0.81$, (**c**) $Fe/SiO_2 = 0.99$ and (**d**) $Fe/SiO_2 = 1.16$.

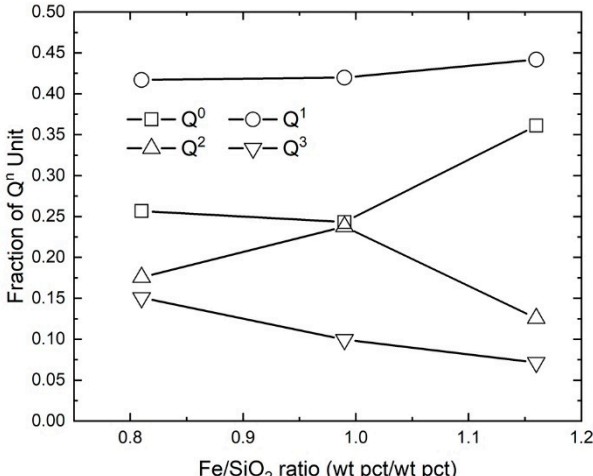

**Figure 4.** The abundance of structural units in FCSM slag as a function of $Fe/SiO_2$ ratio ($pO_2 = 10^{-8}$ atm and $T = 1300\ °C$).

As has been mentioned earlier in the text, the $Q^3/Q^2$ was used as a parameter for the degree of polymerization. This is based on the approach used by previous investigators considering the following reaction in Equation (5) [4,41].

$$[Si_2O_5]^{2-} \leftrightarrow [SiO_3]^{2-} + [SiO_2]. \tag{5}$$

One can write the equilibrium constant (K) of the reaction above, for a given temperature, as

$$K = \frac{[SiO_3]^{2-} \cdot [SiO_2]}{[Si_2O_5]^{2-}} = \frac{Q^2 \cdot Q^4}{Q^3}. \tag{6}$$

By assuming that the equilibrium constant (K) is independent of composition, one can see that the abundance of highly polymerized $Q^4$ units is in proportion to the ratio of $Q^3$ to $Q^2$.

$$\frac{Q^3}{Q^2} \propto \text{Degree of polymerization } (Q^4). \tag{7}$$

In Figure 5a, the $Q^3/Q^2$ ratio of FCSM slag (Fe/SiO$_2$ = 0.81–1.16) was plotted as a function of Fe/SiO$_2$. The figure shows that the $Q^3/Q^2$ of the slag decreases with increasing Fe/SiO$_2$ from 0.81 to 1.16. It should be noted that the slags in the current study were prepared at particular $pO_2$ and contained Fe in the form of FeO and Fe$_2$O$_3$. An appropriate wet bulk chemical analysis is required to determine the actual concentration of FeO (Fe$^{2+}$) and Fe$_2$O$_3$ (Fe$^{3+}$), and their ratio. This analysis was not carried out; rather, the bulk slag chemistry (total Fe) was determined and analysed using ICP-AES. In this situation, the concentrations of FeO and Fe$_2$O$_3$, were determined through equilibrium re-calculations using thermochemical package FactSage 7.2 [42], considering the bulk composition obtained from ICP-AES analysis (in Table 1). FToxid database was used for the FactSage calculations. The concentration of FeO and the Fe$^{3+}$/(Fe$^{2+}$ + Fe$^{3+}$) ratio are also presented in Figure 5a. It was found that both Fe$^{2+}$ and Fe$^{3+}$ ions increase with increasing Fe/SiO$_2$ ratio. The ratio of Fe$^{3+}$/(Fe$^{2+}$ + Fe$^{3+}$) was also found to increase in this range.

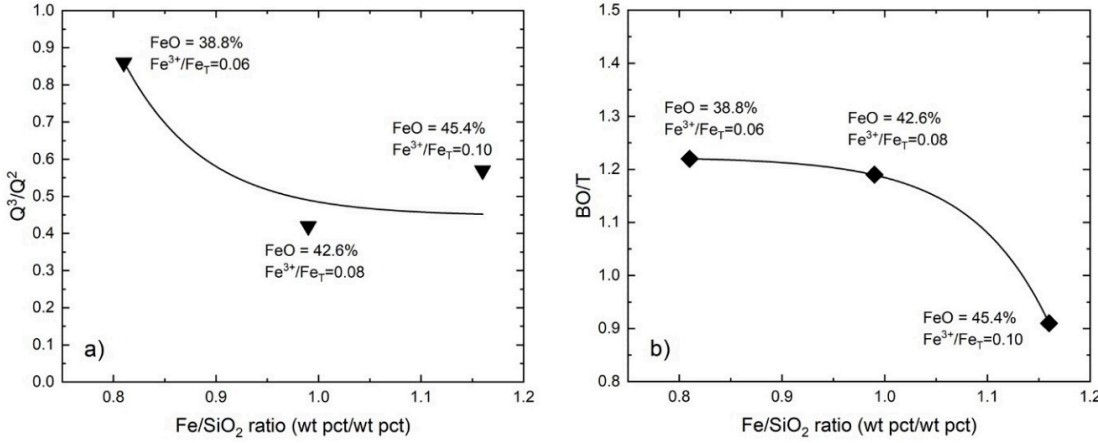

**Figure 5.** Relationship between Fe/SiO$_2$ ratio and (**a**) $Q^3/Q^2$ ratio and (**b**) bridging oxygen per tetrahedral (BO/T) of FCSM slags ($pO_2$ = 10$^{-8}$ atm and $T$ = 1300 °C).

Fe$^{2+}$ and Fe$^{3+}$ cations can essentially modify the structure by incorporating themselves into the tetrahedral position in the silicate network units [36,39]. The ion-oxygen bond strength of Fe$^{3+}$ (Z/$r^2$ = 9.9) is almost twice of Fe$^{2+}$ (Z/$r^2$ = 5.4). Therefore, Fe$^{3+}$ can adopt both four-fold (IV) and six-fold (VI) coordination. Four-fold coordination suggests cations will work as a network former, while six-fold coordination suggests the role of a network breaker. It was reported that the role of Fe$^{3+}$ depends on particularly Fe$^{3+}$/(Fe$^{2+}$ + Fe$^{3+}$) ratio in the slag [5,37,40]. Fe$^{3+}$ acts as a network former when the ratio is >0.5 and acts as a network breaker when the ratio is <0.3. Figure 5a shows that Fe$^{2+}$ and Fe$^{3+}$ ions' concentrations and the Fe$^{3+}$/(Fe$^{2+}$ + Fe$^{3+}$) ratio increase with increasing Fe/SiO$_2$ ratio. Nevertheless, the Fe$^{3+}$/(Fe$^{2+}$ + Fe$^{3+}$) values were found to be less than 0.3, as shown in Figure 5a, which suggests that Fe$^{3+}$ and the Fe$^{2+}$ contribute as network breakers in the slag, as opposed to network formers. Therefore, increasing the Fe/SiO$_2$ ratio resulted in a more depolymerized FCSM slag.

The average bridging oxygen in the silicate network (BO/T) was also calculated from the abundance of different silicate units using Equations (2) and (3). This parameter can also be used to characterize the degree of depolymerization of the silicate melt. As can be seen in Figure 5b, the BO/T was found to decrease with increasing $Fe/SiO_2$ ratio, similarly as $Q^3/Q^2$. This also supports the role of $Fe^{3+}$ as a network breaker in the range of conditions studied.

### 4.1.2. The Effect of Basicity

The FTIR spectra with the wavenumbers ranging from 700 to 1500 $cm^{-1}$ from the FCSM slags of different basicity are shown in Figure 6a. Here, the basicity (B) is defined as (wt.% CaO + wt.% MgO)/wt.% $SiO_2$. As can be seen from Figure 6a, the relative intensity of high frequency (1100 ± 10 $cm^{-1}$) $Q^3$ peak decreases in height with increasing basicity from 0.49 to 0.89. It also shows a weak shoulder in the main silicate envelope at a basicity of 0.89. Similar to the case for different $Fe/SiO_2$, the depth of low frequency (850 ± 5 $cm^{-1}$) peak increases and slightly shifts towards low wavenumber region with increasing basicity.

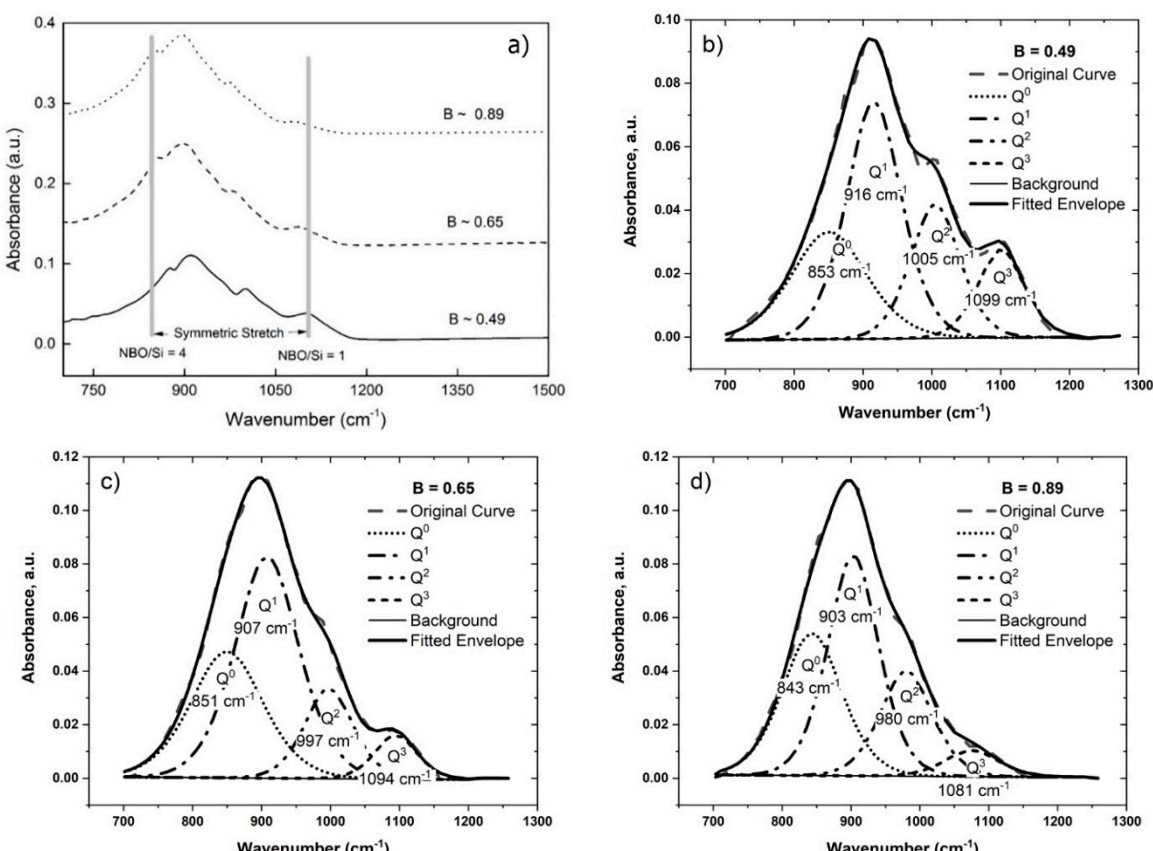

**Figure 6.** (**a**) FTIR spectra of FCSM slags of basicity 0.49 to 0.89 ($pO_2 = 10^{-8}$ atm and $T = 1300\ °C$); and deconvolution results of FTIR spectra for slag with (**b**) basicity (B) = 0.49, (**c**) B = 0.65 and (**d**) B = 0.89.

The FTIR spectra in Figure 6a were then deconvoluted using Gaussian function, and the results are shown in Figure 6b–d. The peak position and relative abundance of the peaks corresponding to different structural units are listed in Table 4. Figure 6b–d shows the envelope of $Q^3$ and $Q^2$ silicates units in the FCSM slag decreases, while $Q^0$ increases with increasing of basicity.

**Table 4.** FTIR peak shift and area fraction from FTIR spectra of FCSM slags of different basicity.

| Sample | Peak Position (cm$^{-1}$) | | | | | Fraction of Different Q$^n$ Units | | | | Q$^3$/Q$^2$ |
|---|---|---|---|---|---|---|---|---|---|---|
| | Basicity | Q$^0$ | Q$^1$ | Q$^2$ | Q$^3$ | Q$^0$ | Q$^1$ | Q$^2$ | Q$^3$ | |
| #3 | 0.49 | 853 | 916 | 1005 | 1099 | 0.26 | 0.40 | 0.22 | 0.12 | 0.53 |
| #4 | 0.65 | 851 | 907 | 997 | 1094 | 0.32 | 0.47 | 0.16 | 0.05 | 0.33 |
| #5 | 0.89 | 843 | 903 | 980 | 1081 | 0.32 | 0.42 | 0.21 | 0.05 | 0.26 |

The relative fractions of the silicate units as a function of basicity are shown in Figure 7. The Q$^3$ and Q$^2$ unit fractions were found to decrease with increasing basicity in the FCSM slag from 0.49 to 0.65. At the same time, Q$^0$ and Q$^1$ were found to increase in this range of basicity, indicating transformation of Q$^3$ and Q$^2$ to depolymerized Q$^0$ and Q$^1$. Increasing basicity from 0.65 to 0.89 resulted in a decrease of Q$^1$, while Q$^0$ and Q$^3$ remained unchanged. It was also observed that in this basicity range, Q$^2$ was increased. Therefore, in this range, it appeared that the main transformation was from Q$^1$ to Q$^2$.

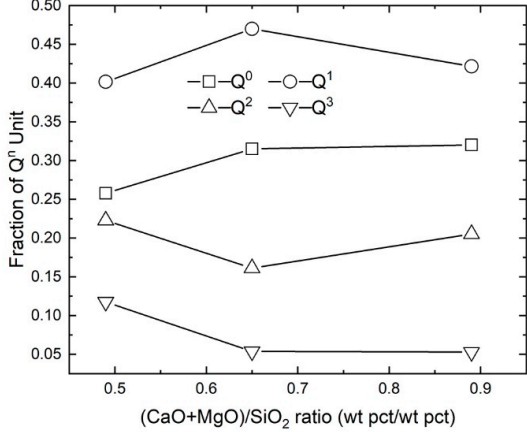

**Figure 7.** The abundance of structural units in FCSM slag as a function of basicity ($pO_2 = 10^{-8}$ atm and $T = 1300\ °C$).

Figure 8 shows the relationship between of Q$^3$/Q$^2$ of the FCSM slags with basicity in the basicity range of 0.49–0.89. The Q$^3$/Q$^2$ was observed to decrease with increasing basicity, which suggests that both Ca$^{2+}$ and Mg$^{2+}$ cations act as network modifiers in the current slag system, and thus increase the depolymerization. The parameter Q$^3$/Q$^2$ is directly relevant to basicity. In fact, the Q$^3$/Q$^2$ may be more suitable parameter to represent "basicity" of the slag compared to the conventional ratio of basic and acidic oxides [(CaO + MgO)/SiO$_2$], as it represents directly the structure of the slag which can also be directly measured.

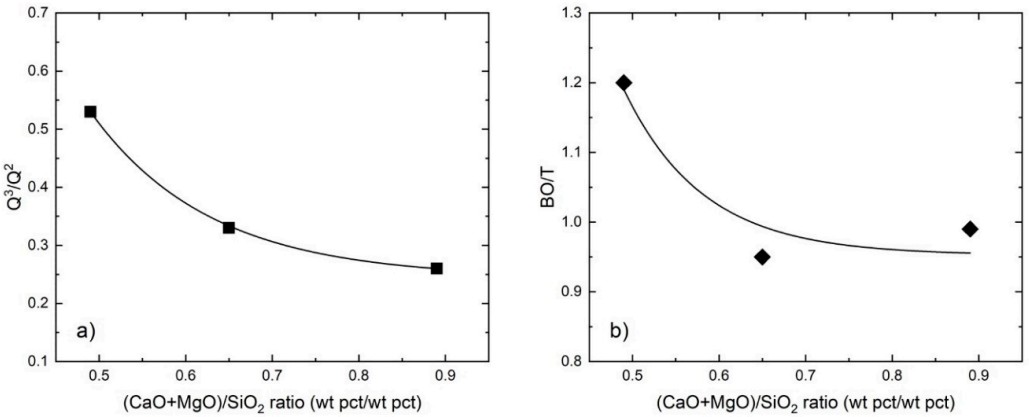

**Figure 8.** Relationship between basicity and (**a**) Q$^3$/Q$^2$ ratio; and (**b**) bridging oxygen per tetrahedral (BO/T) of FCSM slags ($pO_2 = 10^{-8}$ atm and $T = 1300\ °C$).

The ion-oxygen bond strength of $Mg^{2+}$ ($Z/r^2$ = 3.9) is much higher than $Ca^{2+}$ ($Z/r^2$ = 2), and the ability of MgO to offer $O^{2-}$ is weaker than CaO. Thus, the polarizing power of $Mg^{2+}$ is stronger than that of $Ca^{2+}$, and $Mg^{2+}$ has a strong electrostatic interaction with another ion in the presence of $Ca^{2+}$ in slag melts. This suggests that CaO has a higher ability to act as network modifier than MgO in the FCSM slag system.

The parameter of average bridging oxygen, which can also be used to characterize the degree of depolymerization of silicate melt, was calculated from the abundance of the Si–O structural units using Equations (2) and (3). It is generally known that the degree of polymerization should decrease with basicity, which could be confirmed by the decrease of average bridging oxygen in the silicate slags. The decreasing trend of average bridging oxygen (BO) with increasing slag basicity is shown in Figure 8b.

### 4.2. Structure Analysis of the FeOx-CaO-SiO₂-MgO-Cu₂O-PdO (FCSM-Cu₂O-PdO) Slags

#### 4.2.1. The Effect of Oxygen Potential and Temperature

The FTIR analyses results of the FCSM-$Cu_2O$-PdO slags are given in Figure 9a,b for frequencies ranging from 700 to 1500 $cm^{-1}$. The FTIR spectra obtained from the FCSM-$Cu_2O$-PdO slags, equilibrated at different temperatures and $pO_2$, were deconvoluted to quantify the relative fraction of different silicate units and selected results are plotted Figure 10a–d. The detailed values of the fraction of silicate units are presented in Table 5.

**Table 5.** FTIR peak shift and area fraction obtained from FTIR spectra of FCSM-$Cu_2O$-PdO slags equilibrated under different temperature and $pO_2$ conditions.

| Sample | T (°C) | log $pO_2$ | Peak Position ($cm^{-1}$) | | | | Fraction of Different $Q^n$ Units | | | | $Q^3/Q^2$ |
|---|---|---|---|---|---|---|---|---|---|---|---|
| | | | $Q^0$ | $Q^1$ | $Q^2$ | $Q^3$ | $Q^0$ | $Q^1$ | $Q^2$ | $Q^3$ | |
| #4-b | 1300 | −9 | 850 | 914 | 999 | 1102 | 0.29 | 0.45 | 0.20 | 0.06 | 0.27 |
| #4-c | 1300 | −10 | 850 | 911 | 998 | 1102 | 0.29 | 0.41 | 0.23 | 0.07 | 0.29 |
| #2-a | 1200 | −8 | 850 | 911 | 1004 | 1100 | 0.29 | 0.45 | 0.18 | 0.07 | 0.40 |
| #2-c | 1300 | −8 | 845 | 907 | 992 | 1092 | 0.32 | 0.43 | 0.20 | 0.05 | 0.27 |

It can be seen from Figure 9a that the spectra do not show any significant changes with decreasing $pO_2$. That indicates that in the $pO_2$ range studied, changing the $pO_2$ is not enough to significantly change the structure of the slag; i.e., affect the depolymerization. Further analyses were carried out using FactSage 7.2 to evaluate the phase changes in the slags. The results of equilibrium calculations showed that while the concentration of FeO in the slag is relative unchanged (≈40–41 wt.%), there was a steady decrease of the concentration of $Fe_2O_3$ when the $pO_2$ was decreased from $10^{-8}$ atm to $10^{-10}$ atm; i.e., 3.68 wt.%, 2.55 wt.%, 1.44 wt.% at $pO_2$ of $10^{-8}$, $10^{-9}$ and $10^{-10}$ atm, respectively. At $pO_2$ of $10^{-9}$ and $10^{-10}$ atm, it was also calculated that olivine phase (Fe, Mg, Ca)$_2$SiO$_4$ starts to crystallize. Therefore, it appeared that while the general slag structure does not change significantly, reducing the $pO_2$ (or lowering the availability of $O^{2-}$) results in the decrease of $Fe^{3+}$ concentration in the slag and promoting the crystallization of olivine phase.

Figure 9b shows the effect of temperature on the structure equilibrated FCSM-$Cu_2O$-PdO slags. Both of $Q^3$ and $Q^2$ peaks slightly shift toward lower values with increasing temperature. The peak near ≈910 $cm^{-1}$ ($Q^1$) does not shift significantly. The increasing of intensity and the shifting of the absorbance bands towards lower frequency suggest a depolymerization of the slag [43]. The higher temperature increases the kinetic energy of atoms, making it easier to break the tetrahedral units.

The deconvoluted spectra of some selected samples are shown in Figure 10. It can be seen from Table 5 that there were no significant differences on the $Q^3/Q^2$ ratio when the $pO_2$ was decreased from $10^{-8}$ to $10^{-10}$ atm at 1300 °C. At the same time, a significant increase in $Q^3/Q^2$ ratio was observed with decreasing $pO_2$ temperature.

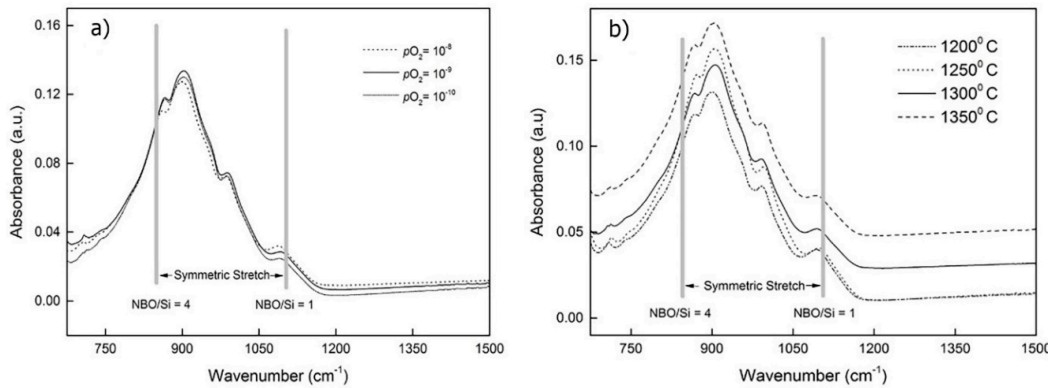

**Figure 9.** FTIR spectra of equilibrated FCSM-Cu$_2$O-PdO slags (**a**) at different $pO_2$ at 1300 °C and (**b**) at different temperatures at $pO_2 = 10^{-8}$ atm.

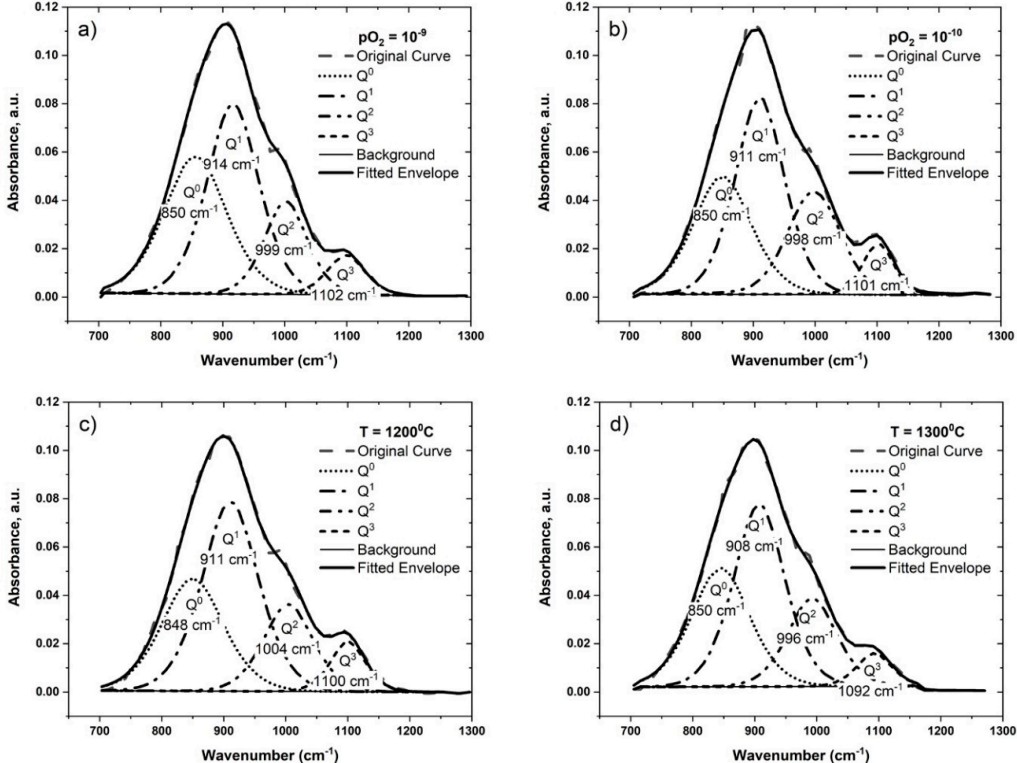

**Figure 10.** Deconvolution results of FTIR spectra of FCSM-Cu$_2$O-PdO slags for samples equilibrated at (**a**) $pO_2 = 10^{-9}$, $T = 1300$ °C (1573 K); (**b**) $pO_2 = 10^{-10}$, $T = 1300$ °C (1573 K); (**c**) $pO_2 = 10^{-8}$, $T = 1200$ °C (1473 K); and (**d**) $pO_2 = 10^{-8}$, $T = 1300$ °C (1573 K).

The structure of the FCSM-Cu$_2$O-PdO slags were correlated with the Pd partition ratio, $L_{Pd}^{\frac{s}{m}}$, at different conditions ($pO_2$ and temperature). Figure 11a shows the change in Q$^3$/Q$^2$ of the FCSM-Cu$_2$O-PdO with $L_{Pd}^{\frac{s}{m}}$ and $pO_2$ at 1300 °C (1573 K). Increasing the $pO_2$ from $10^{-10}$ to $10^{-9}$ atm, did not appear to increase the $L_{Pd}^{\frac{s}{m}}$ significantly; although increasing to this oxygen potential was accompanied by a decrease of Q$^3$/Q$^2$. Further increase in $pO_2$ to $10^{-8}$ atm, however, increased the $L_{Pd}^{\frac{s}{m}}$ significantly. This may suggest that at this $pO_2$ range, there was enough oxygen in the gas phase that allowed the incorporation of the oxygen atom into the silicate network units to form new bonds.

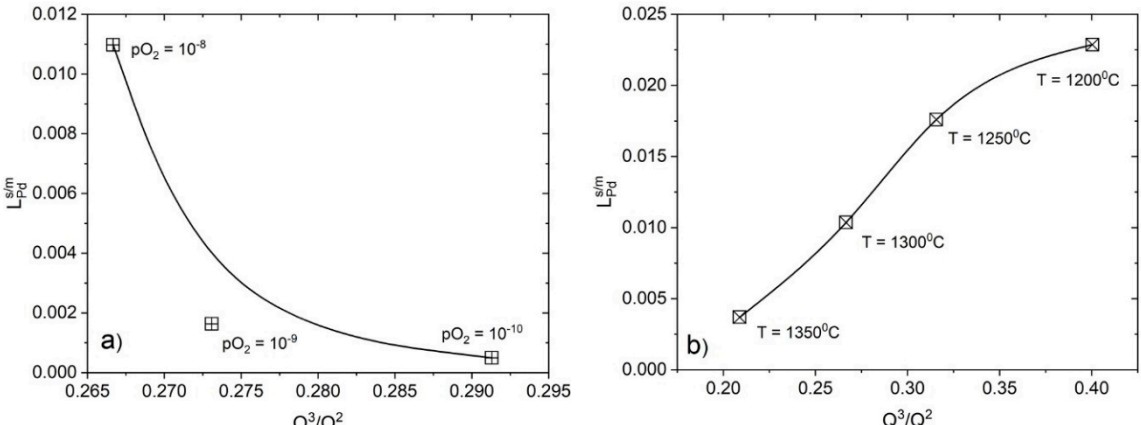

**Figure 11.** Relationship of the $Q^3/Q^2$ and Pd partition-ratio ($L_{Pd}^{\frac{s}{m}}$) in FCSM-Cu$_2$O-PdO slags at: (**a**) fixed temperature of 1300 °C (1573 K) and at different $pO_2$; (**b**) fixed $pO_2$ of $10^{-8}$ atm and at different temperatures.

Figure 11b shows the relationship between the $Q^3/Q^2$, $L_{Pd}^{\frac{s}{m}}$ and temperature at $pO_2 = 10^{-8}$ atm. Both the $Q^3/Q^2$ and the $L_{Pd}^{\frac{s}{m}}$ were found to increase with decreasing temperature from 1350 °C (1623 K) to 1200 °C (1473 K). As mentioned earlier in the text, higher temperature promotes the breakage of the silicate network units that modify the structure. At the same time, the stability of PdO in the slag is less at higher temperature. A similar finding of higher activity co-efficient of PdO in slag at higher temperatures was reported in a previous study [23]. PdO is mildly basic, so less acidic slag (low $Q^3/Q^2$); therefore, a reducing atmosphere allows more Pd to report to the metal phase.

### 4.2.2. The Effect of Slag Composition

The FTIR results of the equilibrated FCSM-Cu$_2$O-PdO slag samples of various compositions (represented by Fe/SiO$_2$ and B ratios) were plotted as a function of the wavenumbers from 700 to 1500 cm$^{-1}$, and they are shown in Figure 12a,b. The detailed data of peak positions and fraction of $Q^n$ units after deconvolution can be seen in the Table 6. As can be seen from Figure 12a, the relative intensity of $Q^2$ and $Q^3$ peaks decreases with increasing Fe/SiO$_2$. The peak of $Q^0$ moves as the Fe/SiO$_2$ is increased. The effect of basicity can be seen in Figure 12b. The relative intensity of $Q^1$ increases and $Q^3$ decreases with increasing basicity. The peaks of $Q^0$, $Q^1$ and $Q^2$ shift significantly with increasing basicity.

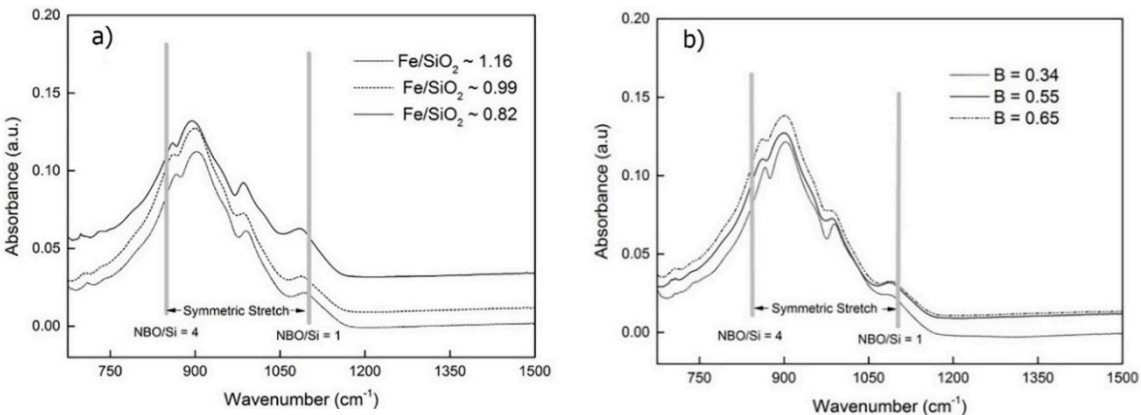

**Figure 12.** FTIR spectra of FCSM-Cu$_2$O-PdO slags equilibrated with Cu ($pO_2 = 10^{-8}$ atm and $T = 1300$ °C (1573 K)) at different: (**a**) Fe/SiO$_2$ ratios and (**b**) levels of basicity.

**Table 6.** FTIR peak shift and area fraction obtained from FTIR spectra of FCSM-$Cu_2O$-PdO slags of different basicity and Fe/$SiO_2$ ratios.

| Sample | Fe/$SiO_2$ | Basicity, B | Peak Position ($cm^{-1}$) | | | | Fraction of Different $Q^n$ Units | | | | $Q^3/Q^2$ |
|---|---|---|---|---|---|---|---|---|---|---|---|
| | | | $Q^0$ | $Q^1$ | $Q^2$ | $Q^3$ | $Q^0$ | $Q^1$ | $Q^2$ | $Q^3$ | |
| #1-a | 0.81 | 0.49 | 845 | 904 | 995 | 1090 | 0.26 | 0.40 | 0.23 | 0.11 | 0.49 |
| #2-c | 0.99 | 0.49 | 840 | 908 | 997 | 1091 | 0.25 | 0.54 | 0.15 | 0.07 | 0.46 |
| #3-a | 1.16 | 0.49 | 845 | 911 | 991 | 1093 | 0.29 | 0.39 | 0.24 | 0.08 | 0.33 |
| #6 | 1.06 | 0.34 | 847 | 909 | 995 | 1091 | 0.27 | 0.41 | 0.24 | 0.07 | 0.29 |
| #2-c | 0.99 | 0.55 | 840 | 905 | 990 | 1095 | 0.28 | 0.44 | 0.21 | 0.07 | 0.31 |
| #4-d | 0.99 | 0.65 | 842 | 905 | 988 | 1099 | 0.30 | 0.40 | 0.25 | 0.05 | 0.19 |

The silicate units' relative fractions in the FCSM-$Cu_2O$-PdO slags were plotted with respect to Fe/$SiO_2$ and basicity, and the results are shown in Figure 13a,b. It can be noticed that silicates structure mainly transformed from highly polymerized $Q^3$ and $Q^2$ units into depolymerized $Q^1$ units, as the Fe/$SiO_2$ is increased from 0.82 to 0.99, as seen in Figure 13a. Beyond this, however, the $Q^1$ appeared to re-transform to $Q^2$ and $Q^0$. The effect of basicity is shown in Figure 13b. It appeared that increasing basicity from 0.49 to 0.65 resulted in the transformation of $Q^2$ to $Q^1$ units. However, further increase of basicity from 0.65 to 0.89 transformed the $Q^1$ back to $Q^2$. There were slight increases and decreases of $Q^0$ and $Q^3$, respectively, as the basicity was increased from 0.49 to 0.89.

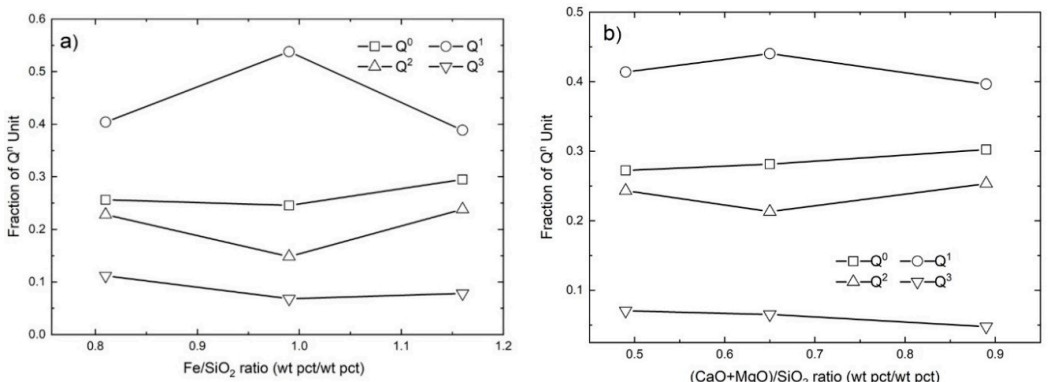

**Figure 13.** The relative abundance of structural units in FCSM-$Cu_2O$-PdO slags equilibrated with Cu ($pO_2 = 10^{-8}$ atm and $T$ = 1300 °C (1573 K)) as a function of: (**a**) Fe/$SiO_2$ ratio, and (**b**) basicity.

The effect of the slag chemistry (Fe/$SiO_2$ and basicity) on the $Q^3/Q^2$ and the Pd partition ratio, $L_{Pd}^{\frac{s}{m}}$, of the FCSM-$Cu_2O$-PdO, was investigated. Figure 14a shows that the $Q^3/Q^2$ increases with decreasing Fe/$SiO_2$ ratio. At the same time, $L_{Pd}^{\frac{s}{m}}$ increases with increasing $Q^3/Q^2$. It appears that the number of $Pd^{2+}$ cations acting as network modifiers of the ionization potential ($Z/r^2$ = 5.4) is quite small. The interrelationship between the basicity, the $Q^3/Q^2$ and the $L_{Pd}^{\frac{s}{m}}$ is shown in Figure 14b. It can be seen from Figure 14b that the $L_{Pd}^{\frac{s}{m}}$ increases with decreasing of basicity; however, the variation with the $Q^3/Q^2$ is not that clear. Initially, the $Q^3/Q^2$ increases with increasing basicity from 0.34 to 0.55, but then decreases as the basicity is further increased to 0.65. In general, basic slag is better for Pd partitioning to the metal.

Within the range of the parameters studied, it was found that decreasing $pO_2$ and increasing temperature resulted in maximizing the partitioning of Pd in the metal phase. At higher temperature, the loss of copper in the slag increases; therefore, copper-smelting operation temperatures in between 1573 and 1623 K is suggested [44]. Increasing Fe/$SiO_2$ and basicity is also favorable for the partitioning of Pd in molten copper. However, the optimum condition for recovering valuable elements for the industries is far more complex, as there could be cut-offs for the recoveries of other, different elements. For example, our previous study showed that increasing basicity from 0.3 to 0.9 results in higher partitioning of Pd in copper phase; however, the partitioning of Ge in copper reduced. Interestingly,

the comparison of total recovery of the elements during smelting showed that the effect of basicity is more prominent in Ge-recovery than Pd-recovery. Therefore, a lower basicity or acidic slag is suggested for optimum recovery of Ge and Pd during copper-smelting [44].

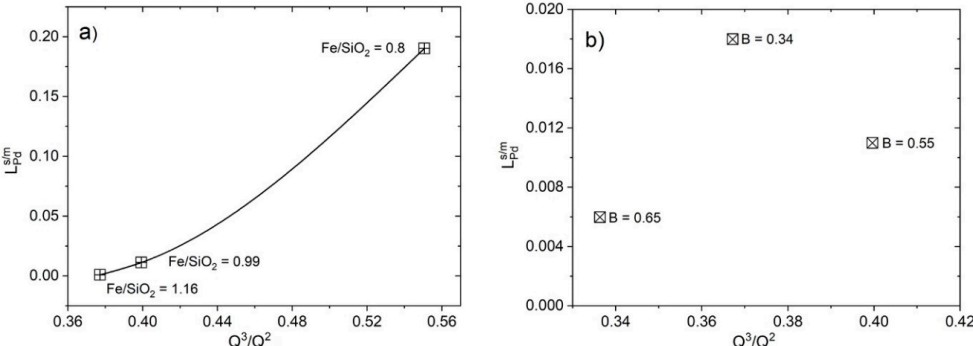

**Figure 14.** Relationship of the $Q^3/Q^2$ and Pd partition-$L_{Pd}^{\frac{s}{m}}$, in FCSM-$Cu_2O$-PdO slags at $pO_2 = 10^{-8}$ atm and $T = 1300$ °C (1573 K) at different: (**a**) Fe/$SiO_2$ ratios and (**b**) levels of basicity.

### 4.3. Correlations of the Slag Structure, with Composition and Experimental Parameters

A correlation that relates the slag structure, composition and experimental parameters has been developed using the broader data from the current and previous studies [33,37] using slag system FCSM, FCSM-$Cu_2O$-PdO and FCSM-$Cu_2O$-$GeO_2$. In this context, the slag structure is expressed using $Q_3/Q_2$, which also represents the degree of polymerization; and correlated with the temperature (T), oxygen partial pressure ($pO_2$) and the compositions of the slags. The correlation, for the general FCSM slag, was developed using a multiple regression analysis using more accurate deconvoluted FTIR spectral data, and it is shown in Equation (8):

$$\frac{Q^3}{Q^2} = \frac{9}{97}\log p_{O_2} - \frac{49713}{T} - \frac{544}{79}\%MgO + \frac{17}{13}\%CaO - \frac{19}{14}\%SiO_2 - \frac{9}{25}\%Fe_T + 123. \tag{8}$$

The percentages of $GeO_2$, PdO and $Cu_2O$ were not included in Equation (8) because of their small sizes. The values of $Q^3/Q^2$ calculated from Equation (8) were compared with the measured values, as shown in Figure 15a,b for the FCSM-$Cu_2O$-$GeO_2$ and FCSM-$Cu_2O$-PdO slags, respectively. Very good correlations ($R^2 = 0.90$ for Ge and $R^2 = 0.86$ for Pd) were found between the calculated and measured values. This is better for the FCSM-$Cu_2O$-$GeO_2$ system found in earlier study, where coefficient of determination, $R^2$ was 0.81 [33]. This equation appears to be sufficient for determining the degree of polymerization (as represented by $Q^3/Q^2$) of the FCSM based slags (containing small amount of other oxides) at specific temperature, $pO_2$ and slag composition.

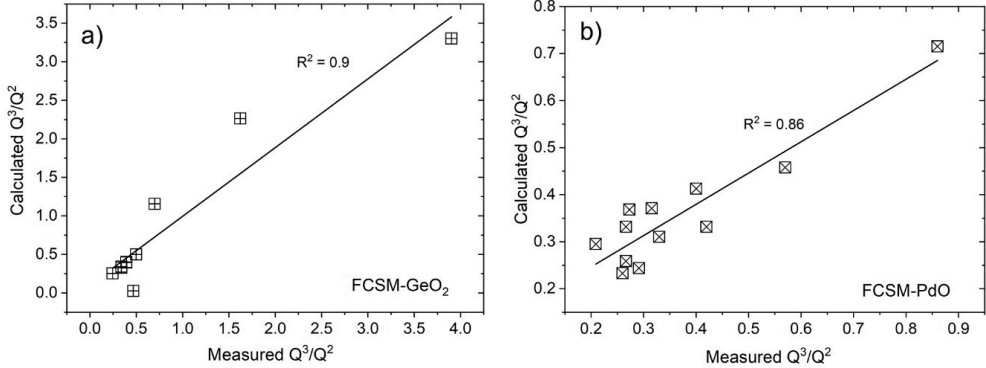

**Figure 15.** Comparison of calculated and measured values of the $Q^3/Q^2$ for (**a**) FCSM-$Cu_2O$-$GeO_2$ and (**b**) FCSM-$Cu_2O$-PdO slags.

It was reported that the Gibbs free energy and thermodynamic properties can be correlated to the slag structure units [3]. In the present study, relationships between the partition-ratio of Ge and Pd, and the degree of polymerization, were re-established using the linear regression analysis and are presented in Equations (9) and (10).

$$L_{Ge}^{S/m} = \frac{5198}{84} \log p_{O_2} - \frac{15004}{T} - \frac{5}{16} \frac{Q^3}{Q^2} + \frac{5198}{99}. \tag{9}$$

$$L_{Pd}^{S/m} = -\frac{1}{910} \log p_{O_2} - \frac{970}{T} + \frac{15}{31} \frac{Q^3}{Q^2} + \frac{15}{32}. \tag{10}$$

The calculated partition ratio values from Equations (9) and (10) were also compared with the relevant measured values, as shown in Figure 16a,b, respectively. The correlations were found to be very good for Ge-containing slag and quite reasonable for Pd-containing slag, $R^2 = 0.92$ and $R^2 = 0.796$, respectively. Equation (9) is more accurate compared to the previous iteration [33]. These equations, therefore, may be used to estimate the $L_{X=(Ge,Pd)}^{\frac{s}{m}}$ of relevant slag system at a specific temperature, $pO_2$ and slag structure. Equations (8)–(10) provide the platform to calculate and reasonably predict the structures of silicate slags and the partitioning ratios for Ge and Pd at the processing conditions.

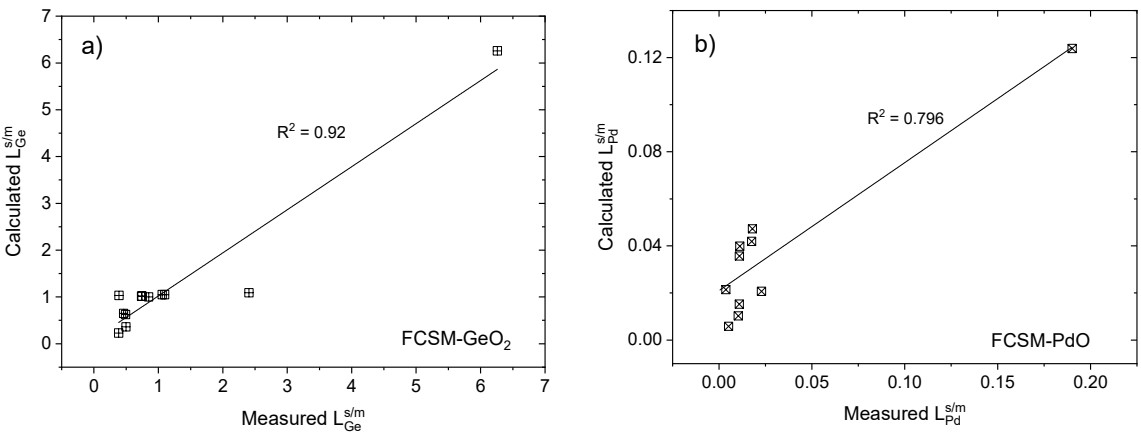

**Figure 16.** Comparison of calculated and measured values of the partitioning ratios of (**a**) Ge and (**b**) Pd in the FCSM slag.

Similar equations for the various valuable elements (Au, Ag, Pt, In, Ru, etc.) should also be developed. These sets of equations will be useful for evaluating and designing slags to be used for secondary black copper process for an optimized process with maximum recovery of all valuable metals.

Following the work of Mysen et al. [3,39,40,43,45], and noting that, $Q^0 = [SiO_4]^{4-}$, $Q^1 = [Si_2O_7]^{6-}$, $Q^2 = [SiO_3]^{2-}$ and $Q^3 = [Si_2O_5]^{2-}$, it is possible to write different possible stoichiometric reactions between the cations present in the slag and the silicate units, as shown in Equations (11)–(34).

$$2Ca^{2+} + Q^0 = Ca_2Q^0; \tag{11}$$

$$3Ca^{2+} + Q^1 = Ca_3Q^1; \tag{12}$$

$$6Ca^{2+} + 3Q^2 = Ca_6Q^2_3; \tag{13}$$

$$Ca^{2+} + Q^3 = CaQ^3; \tag{14}$$

$$2Mg^{2+} + Q^0 = Mg_2Q^0; \tag{15}$$

$$3Mg^{2+} + Q^1 = Mg_3Q^1; \tag{16}$$

$$6Mg^{2+} + 3Q^2 = Mg_6Q^2_3; \tag{17}$$

$$Mg^{2+} + Q^3 = MgQ^3; \tag{18}$$

$$2Fe^{2+} + Q^0 = Fe_2Q^0; \tag{19}$$

$$3Fe^{2+} + Q^1 = Fe_3Q^1; \tag{20}$$

$$6Fe^{2+} + 3Q^2 = Fe_6Q^2_3; \tag{21}$$

$$Fe^{2+} + Q^3 = FeQ^3; \tag{22}$$

$$4Fe^{3+} + 3Q^0 = Fe_4Q^0_3; \tag{23}$$

$$2Fe^{3+} + Q^1 = Fe_2Q^1; \tag{24}$$

$$4Fe^{3+} + 3Q^2 = Fe_4Q^2_3; \tag{25}$$

$$2Fe^{3+} + 3Q^3 = Fe_2Q^3_3; \tag{26}$$

$$Ge^{4+} + Q^0 = GeQ^0; \tag{27}$$

$$3Ge^{4+} + 2Q^1 = Ge_3Q^1_2; \tag{28}$$

$$Ge^{4+} + Q^2 = GeQ^2; \tag{29}$$

$$Ge^{4+} + 2Q^3 = GeQ^2_3; \tag{30}$$

$$2Pd^{2+} + Q^0 = Pd_2Q^0; \tag{31}$$

$$3Pd^{2+} + Q^1 = Pd_3Q^1; \tag{32}$$

$$6Pd^{2+} + 3Q^2 = Pd_6Q^2_3; \tag{33}$$

$$Pd^{2+} + Q^3 = PdQ^3. \tag{34}$$

The Gibbs free energy values of reactions of the above can be calculated considering an ideal mixing which has been successfully used by Halter and Mysen [3] for an ideally mixed $Na_2O$-$SiO_2$ binary system. They reported that the thermodynamic properties of the species can be determined from speciation data collected by the spectroscopic technique; those include enthalpy, heat capacity and entropy. As the slag becomes more complex, the behavior will deviate from ideal mixing. Further modification of the approach, such as using a quasichemical solution model, can also be implemented for a more complex system. Therefore, it should be possible to derive more fundamental relationships between $Q^3/Q^2$ and the thermodynamics of the system by looking at the thermodynamics of the individual reactions between polymeric species and individual cations, as above.

The concept of using $Q^3/Q^2$ as a measure of basicity and correlating this ratio to properties and thermodynamic relationships looks promising but needs further evidence and data to support the claims being made in this paper. There are both theoretical and experimental challenges to establishing $Q^3/Q^2$ as a measure of "basicity." We hope that the current study will stimulate further work in the area.

## 5. Summary

In this work, the structures of the FCSM and FCSM-$Cu_2O$-PdO slags in conditions relevant to black copper smelting operations have been studied using FTIR spectroscopy. The FTIR spectra of the slags were measured to investigate the effects of temperature, $pO^2$ and slag composition on the silicate structures from a broader set of experimental data. The structures of the slags were interpreted based on the abundance of silicon tetrahedra units ($Q^n$). The conclusions drawn from the study include:

- The relative intensity of high-frequency band $Q^3$ to the low-frequency band $Q^0$ in each slag was affected by Fe/$SiO_2$ ratio, basicity, oxygen partial pressure and temperature.
- The degree of polymerization, represented by $Q^3/Q^2$, and the average number of bridging oxygens (BO/T) were found to decrease with increasing both Fe/$SiO_2$ ratio and basicity.

- There are interrelationships between the partition ratio $L_{Pd}^{\frac{s}{m}}$, the $Q^3/Q^2$ and the slag chemistry, which are affected by the experimental conditions; e.g., temperature and $pO_2$.
- Semi-empirical equations have been developed using multiple regression analysis to correlate the $Q^3/Q^2$, temperature, $pO_2$ and slag compositions. These correlations can be used to predict the $Q^3/Q^2$ of the FCSM slag, within the range of compositions investigated.
- The partition ratios of Pd and Ge can be reasonably correlated with the $pO_2$, temperature and $Q^3/Q^2$. Similar equations for the various valuable elements (Au, Ag, Pt, In, Ru, etc.) should also be developed. These sets of equations will be useful for evaluating and designing slags to be used for secondary black copper process for an optimized process with maximum recovery of all valuable metals.

The key point from the present study is that the degree of polymerization of the silicate slags ($Q^3/Q^2$) can be determined from the chemistry (slag composition), and that this can be correlated with process conditions ($T$, $pO_2$) and the partition ratios of valuable elements. Quantitative studies of slag structure have the potential to provide useful parameters for predicting slag properties and thermodynamics of the reactions. Nevertheless, further studies are required for a more rigorous approach.

**Author Contributions:** M.M.H.: writing—review and editing and formal analysis; M.A.R.: conceptualization, review and editing; M.A.H.S.: writing—original draft preparation and formal analysis; G.A.B.: conceptualization, and review and editing. All authors have read and agreed to the published version of the manuscript.

**Funding:** This study was part of M. Al Hossaini Shuva's doctoral study, which was supported by SUPRA (Swinburne University Postgraduate Research Award) and the Wealth from Waste Research Cluster. The latter is a collaborative research program between the Australian CSIRO (Commonwealth Scientific and Industrial Research Organisation), Swinburne University of Technology, University of Technology Sydney, Monash University, University of Queensland and Yale University, USA.

**Conflicts of Interest:** The authors declare no conflict of interest. The funders had no role in the design of the study; in the collection, analyses, or interpretation of data; in the writing of the manuscript, or in the decision to publish the results.

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
