# Peer review of "Study of the Structure of FeOx-CaO-SiO2-MgO and FeOx-CaO-SiO2-MgO-Cu2O-PdO Slags Relevant to Urban Ores Processing through Cu Smelting"

_metals, doi:10.3390/met10010078_

Round 1

Reviewer 1 Report

Dear authors,

The manuscript “Study of the Structure of FeOx-CaO-SiO2-MgO and FeOx-CaO-SiO2-MgO-Cu2O-PdO Slags Relevant to Urban Ores Processing through Cu Smelting” Hasan et al., investigates the FeOx-CaO-SiO2-MgO and FeOx-CaO-SiO2-MgO-Cu2O-PdO phase systems in respect of their structure and the formed silicon oxide ion species. The study is excellent written and well documented based on various spectroscopic analyses. Therefore, I suggest the publishing of the current work with the addition of few information:

In the introduction part you should mention in which cases (kind of electronic wastes, whether pyrometallurgy for recycling has been applied at laboratory or industrial scale) the FeOx-CaO-SiO2-MgO-Cu2O-PdO can be found and how the knowledge of its structural characteristics will help to the more efficient separation of the metallic values. In Figure 1: The addition of CuO-PdO in the slag system is not clear at which step is taking place. Also, the smelting temperature can be quoted in Figure 1. You mention the use of FactAge for the design of phase diagrams – You can add 1-2 basic phase diagrams in the MS.

It is very important your results somehow correlated to technical information. You can mention for example which are the optimum conditions (temperature, oxides ratios) for the decreasing of the viscosity.

Author Response

Response to Reviewer 1 Comments

Dear reviewer,

 At first thank you very much for reviewing the manuscript “Study of the Structure of FeOx-CaO-SiO2-MgO and FeOx-CaO-SiO2-MgO-Cu2O-PdO Slags Relevant to Urban Ores Processing through Cu Smelting”. Please see the below responses regarding the issues raised.

Point 1: In the introduction part you should mention in which cases (kind of electronic wastes, whether pyrometallurgy for recycling has been applied at laboratory or industrial scale) the FeOx-CaO-SiO2-MgO-Cu2O-PdO can be found and how the knowledge of its structural characteristics will help to the more efficient separation of the metallic values.

Response 1: Thank you very much for the suggestion. The following sentences are added in page 2 to clarify the query:

“Previous research suggests that printed circuit boards (PCBs) contain as much as 220 ppm of Pd [8] which is equivalent to USD 13,747 per ton of circuit boards [9]. As PCBs are rich in copper and other valuable metals, secondary copper smelters, including Boliden Ronnskar, Umicore, Dowa mining, L.S. Nikko and others use PCBs as charge in the process. The inclusions of PdO in the slag is expected during the smelting process as PCBs contains Pd. A thorough study on the effect of process parameters and slag compositions on the partitioning of Pd in the process is necessary to ensure maximum recovery of the metals. Moreover, the effect of PdO on the structure of slag is also important, as the thermo-physical and thermo-dynamic properties of the slags are related to slag structures [10].”

Takanori, A. Ryuichi, M. Youichi, N. Minoru, T. Yasuhiro and A. Takao: Techniques to separate metal from waste printed circuit boards from discarded personal computers. Journal of material cycles and waste management, 2009, vol. 11, pp. 42-54. Palladium Spot Price History & Current Prices, in Money Metal Exchange (https://www.moneymetals.com/precious-metals-charts/palladium-price). Accessed on 18 December 2019. A. Brooks, M.M. Hasan and M.A. Rhamdhani: Slag Basicity: What Does It Mean? Jiang T. et al. (eds) 10th International Symposium on High-Temperature Metallurgical Processing. The Minerals, Metals & Materials Series. Springer, Cham, 2019: p. 297-308.

Point 2: In Figure 1: The addition of CuO-PdO in the slag system is not clear at which step is taking place.

Response 2: Thanks for the comment. The following sentences are added in page 4-5 of the MS to explain the addition of Cu2O-PdO. Figure 1 explains the preparation of FCSM master slags only FCSM

“The FCSM-Cu2O-PdO slags were prepared by equilibration of the FCSM slag with the Cu-Pd master-alloy for 6 hours. The thermodynamics of the system at the given temperature and partial pressure of oxygen allowed partial oxidation of the master-alloy and contribute the Cu2O and PdO species in the slag.”

Moreover, please check page 6, line 212-229 for more details of the equilibration study.

Point 3: Also, the smelting temperature can be quoted in Figure 1.

Response 3: Thanks for the suggestion, melting temperature added in the Figure title.

Point 4: You mention the use of FactAge for the design of phase diagrams – You can add 1-2 basic phase diagrams in the MS.

Response 4: Thanks for this suggestion. In this study we estimated the concentration of FeO and Fe2O3, through equilibrium re-calculations using thermochemical package FactSage 7.2 in correlation with the bulk composition obtained from ICP-AES. The calculations are done for various temperatures and partial pressure of oxygen. Incorporating a number of phase diagrams in the MS may divert the main message of the paper and therefore the authors are not comfortable to add the phase diagrams in this paper.  

Point 5: It is very important your results somehow correlated to technical information. You can mention for example which are the optimum conditions (temperature, oxides ratios) for the decreasing of the viscosity.

Response 5: Thank you very much for the suggestion. From this research we have suggested a general condition for the maximum recovery of Pd. The following sentences are added in page 16:

“Within the range of the parameters studied, it is found that decreasing pO2 and increasing temperature results in maximizing the partitioning of Pd in the metal phase. At higher temperature, the loss of copper in the slag increases, therefore, copper-smelting operation temperatures in between 1573 and 1623 K is suggested [45]. A higher Fe/SiO2 and basicity is also found favourable for the partitioning of Pd in molten copper. However, the optimum condition for recovering valuable elements for the industries is far more complex as there could be cut-offs for the recovery of different other elements. For example, our previous study showed, increasing basicity from 0.3 to 0.9 results in higher partitioning of Pd in copper phase, however, the partitioning of Ge in copper reduces. Interestingly, the comparison of total recovery of the elements during smelting showed that the effect of basicity is more prominent in Ge-recovery than Pd-recovery. Therefore, a lower basicity or acidic slag is suggested for optimum recovery of Ge and Pd during copper-smelting [45].”

Analysis for optimum conditions for recovery of valuable metals from e-waste through black copper smelting. in 8th International Symposium on High-Temperature Metallurgical Processing. 2017. Springer, pp. 419-427.

Reviewer 2 Report

The manuscript shows an interesting research about the structure of different slags, with a systematic study of the polymerization of the silicates. I recommend its publication after the correction of a minor typographical mistake: check the symbol of degree Celsius.

Author Response

Dear reviewer,

At first thank you very much for reviewing the manuscript “Study of the Structure of FeOx-CaO-SiO2-MgO and FeOx-CaO-SiO2-MgO-Cu2O-PdO Slags Relevant to Urban Ores Processing through Cu Smelting”. Please see the below responses regarding the issues raised.

Point 1: check the symbol of degree Celsius.

Response 1: Thanks for your comment. The symbol of degree Celsius is checked in the whole document and corrected where necessary.

Reviewer 3 Report

1) Significant number of references given in the present study use the term "distribution" when talking about the ratio of concentrations of minor elements in slag/metal. In the present study a "partition ratio" term is used. If the authors insist on using this term, please give a definition in the text.

2) Table 1 is incomplete: The concentration of Pd in slag and the compositions of Cu-Pb alloy are not given, which makes it more difficult to compare the results of the present study with other sources. 

Reviewer 4 Report

The title is a bit bulky.

The subject is relevant and should be interesting to the readers of the journal.

The abstract is well-written and concise.

The Introduction is good, with clear motivation given for the present work. You use the term “urban ores” in the title and in the introduction. Are there urban ores other than e-waste that are processed by black copper smelting? Maybe you should be more specific and use the term “e-waste” since it is a valuable Cu-containing urban ore. For the sake of the readers, you should explain briefly what e-waste is composed of and the valuable components besides copper it contains such as Au, Ag, Pt, In, Ru, etc. in e-waste as mentioned as in Line 482 and in the summary.

In the introduction, you should add a brief description of which anionic units that Q1 to Q4 This is only described later in Table 2 and Lines 283-288.

Why are FCS and FCSM slags relevant to black copper smelting with application to e-waste?

The literature review is very thorough with relevant and up-to-date citations, although there are several self-citations.

Why was the extra effort taken to calcine CaCO3 instead of using reagent-grade CaO directly? Was this due to purity requirements? You describe the preparation of Cu-Pd alloy, but what about preparation of the Cu-Ge alloy? Were not FSCM-GeO2 slags also prepared in the current study, since data points for this system are presented in Fig. 15a, Fig. 16a, and Eq. 9?

Table 1: PdO content is not shown in latter part of the table.

Can you mention the range of liquidus temperatures of these slag compositions?

Lines 214-222: You mention that several initial experiments were conducted to determine minimum run time to attain equilibrium. Are you referring to the results presented in Fig. 2 of Reference 19?  If so, please cite that reference here.

Lines 173-174: What do you mean by “working thermocouples”? Where were the thermocouples placed in the reaction tube (not shown in Figure 2), and what type of thermocouples were they?  How do you “calibrate” thermocouples?

Line 186: What do you mean “reference thermocopule” (misp.) here?  You are describing an oxygen probe, not a thermocouple.

1: Why were an inner mullite tube and an outer alumina tube used? The schematic is identical to Fig. 1 in Reference 19, except for a magnesia crucible rather than an alumina crucible. Yet that reference describes using a Type R thermocouple to measure temperature rather than “working thermocouples” as you mention.  They also describe the quenching process, which you have not.

Why were magnesia crucibles chosen?

How was the quenching of the slags carried out?

Please cite the reference to “Yazawa and Takeda”, and Eq. 4 as was done in Reference 19. What was the basis for the chosen range of partial pressure of oxygen?

Was it checked to what extent the slag compositions changed after 16 hour runs equilibrated with the prevailing furnace atmosphere with fixed oxygen potential and the magnesia crucible?

Please explain in a bit more detail how “equilibrium re-calculations” were done in Factsage. Line 296: please cite “Factsage 7.2” as requested by Factsage (see their website). Describe which database was used.

Please define the oxidation reaction for Pd and the partition ratio for palladium with equations in the text as Eq. 7 and Eq. 2 respectively in Reference 19. You give a very vague explanation for why the LPd decreases with increasing Q3/Q2 and increases with increasing temperature shown in Figure 11. A more fundamental description of the thermodynamics of the oxidation reaction for Pd would be appropriate. In Ref. 19 with most of the same co-authors as this manuscript, quite a deep analysis was done of the thermodynamics for Pd partitioning. In fact, from this reference you have available activity coefficient data for nearly some of the same slag compositions as in your study.

How do your results correlate with the findings in Reference 19? For example, in Lines 410-411: “At the same time the stability of PdO in the slag is less at higher temperature”. If stated in thermodynamic terms this implies that the activity of PdO in the slag increases with increasing temperature, and therefore the activity coefficient of PdO increases. This statement seems to agree qualitatively with Fig. 6b in Ref 19.

In Figure 11, it seems to me that multiple variables have been changed simultaneously that impact the partition ratio, namely the temperature, the activity coefficient of PdO implicitly through a change in Q3/Q2, and the partial pressure of oxygen.

Line 469: Is that the correct reference?

Figure 16: In the text you describe the plots in Figure 16 as showing the partition ratios for Ge and Pd; however, the plots shown are identical to Figure 15!

No slag structure analyses are provided in the present work for the FCSM-GeO2.

The discussion in Lines 486-503 is thoughtful, but it seems it already may be possible, at least Pd and Ge, to make the bridge between slag structure and thermodynamic properties. Perhaps it is possible with a deeper analysis for example to correlate the activity coefficient of PdO from Reference 19 to the Q3/Q2 ratio in the present work.

The manuscript is quite well written. However, I have identified several minor language corrections and also given some suggestions for small improvements: Line 29: I suggest “Slags play a vital role...” instead of singular. Line 30: I suggest “…and structure of slags control…” Line 31: “… the deportment of selected valuable elements”, though I am not sure “deportment” is the proper word choice here. Line 37: “the designing of suitable slags…” Line 49: “…number of bridging oxygens…” Line 62: “…refractory into the slag occurs.” Line 74: “…wear of the furnace linings.” Line 111: “… a significant amount of Al2O3.” Line 175, 178: “flown” is not the right word choice in this context. Line 184: “...which is shown schematically…” Line 227: “…slag samples broken into pieces…” Line 236-237: “…used in the current study.” Line 268:  add the word “ratio” to the end of the table caption. Line 270-271: “…as a function of increasing Fe/SiO2 ratio…” Line 272:  add the word “ratio” after “Fe/SiO2”. Same in Line 274, 277. There are several instances in the manuscript where “ratio” should be added after “Fe/SiO2“. Line 288: “…Q4 units…” Line 290: “…the Q3/Q2 ratio…” Line 303: “…work as a network former…”.  Similar in Lines 304, 306 and 315. Line 309: “…as opposed to network formers.” Line 320: “…peak decreases in height…” Line 329; “…as a function of basicity are shown in Figure 7.” Line 384: “…thus making it easier to break…” Line 406: “…to form new bonds.” Line 552:  Should be “E.T. Turkdogan”.

Reviewer 5 Report

In this manuscript a study of the structure of FCS-MgO (FCSM) and FCS-MgO-Cu2O-PdO (FCSM-Cu2O-PdO) slags is studied to be used to improve the metal recovery in black copper smelting. The influence of chemical composition, oxygen partial pressure and temperature is analyzed.

The paper is well written and the authors present a good state of art on the subject under discussion. The methodology is correct but it is a shame that the authors only used three different compositions to do the study. Many results are not conclusive because the small number of compositions considered.

I suggest minor changes to improve the manuscript. Here I include some suggestions.

Line 45. This equation was first presented by Toop, G. W., Somis, C. S. (1962). Some new ionic concepts of silicate slags. Canadian Metallurgical Quarterly, 1(2), 129-152.

Line 46. [SiO4]4-

Line 48.The first time write also FTIR in full.

Line 71. Please, could you include a citation?.

Line 145-147. Replace pct or delete pct, eg 99.5% wt pct replace by 99.5 wt.% pure.

Line 150 A mixture was mixed (???). Replace conveniently, eg. Replace mixed by homogenized.

Line 181. A reference is necessary.

Line 193. Change the location of the dot as  inlet [18].

Line 285: [Si2O5]2− ↔ [Si2O6]4− + [SiO2] (?),

The reaction is not matched and if you write [SiO2], this minds that it is completetly polimerized. Then, the reaction should be 2[Si2O5]4− ↔ [Si2O6]4− + 2[SiO4]4-

Round 2

Reviewer 4 Report

Dear Authors,

Thank you for your responses and alterations to the manuscript.

I look forward to the continuation of your research in correlating slag structure with thermodynamic properties.